# Cannabinoid and Opioid Receptor Affinity and Modulation of Cancer-Related Signaling Pathways of Machaeriols and Machaeridiols from *Machaerium* Pers.

**DOI:** 10.3390/molecules28104162

**Published:** 2023-05-18

**Authors:** Ilias Muhammad, Mohammad A. Ibrahim, Mallika Kumarihamy, Janet A. Lambert, Jin Zhang, Marwa H. Mohammad, Shabana I. Khan, David S. Pasco, Premalatha Balachandran

**Affiliations:** 1National Center for Natural Products Research, Research Institute of Pharmaceutical Sciences, School of Pharmacy, The University of Mississippi, University, MS 38677, USA; mmibrahi@olemiss.edu (M.A.I.); mallika.kumarihamy@usda.gov (M.K.); janetlambert@med.unr.edu (J.A.L.); jzhang3@olemiss.edu (J.Z.); skhan@olemiss.edu (S.I.K.); dpasco@olemiss.edu (D.S.P.); 2Department of BioMolecular Sciences, Division of Pharmacognosy, School of Pharmacy, The University of Mississippi, University, MS 38677, USA

**Keywords:** *Machaerium* Pers., machaeriol A–D, machaeridiol A–C, hexahydrodibenzopyran, phytocannabinoid, CB1, CB2, opioid, anticancer, cytotoxicity, transcription factors, cancer signaling pathways

## Abstract

Machaeriols and machaeridiols are unique hexahydrodibenzopyran-type aralkyl phytocannabinoids isolated from *Machaerium* Pers. Earlier studies of machaeriol A (**1**) and B (**2**) did not show any affinity for cannabinoid receptor 1 (CB1 or CNR1), although they are structural analogs of psychoactive hexahydrocannabinol. This study comprehensively reports on the affinities of isolated *Machaerium* Pers. compounds, namely machaeriol A–D (**1**–**4**) and machaeridiol A–C (**5**–**7**), against cannabinoid (CB1 and CB2) and opioid (*κ*, *δ* and *µ*) receptors. Among the isolated compounds, machaeriol D (**4**) and machaeridiol A–C (**5**–**7**) showed some selective binding affinity for the CB2 receptor, using a radioligand binding assay, with *K*_i_ values of >1.3, >1.77, >2.18 and >1.1 μM, respectively. On the other hand, none of the compounds showed any binding to the CB1 receptor. Due to recent reports on the anticancer potential of the endocannabinoid system, compounds **1**–**7** were tested against a battery of luciferase reporter gene vectors that assess the activity of many cancer-related signaling pathways, including Stat3, Smad2/3, AP-1, NF-κB, E2F, Myc, Ets, Notch, FoxO, Wnt, Hedgehog and pTK in HeLa and T98G glioblastoma cells. Complete dose–response curves have been determined for each compound in both of these cell lines, which revealed that machaeridiol **6** displayed activities (IC_50_ in µM in HeLa and T98G cells) towards Stat3 (4.7, 1.4), Smad2/3 (1.2, 3.0), AP-1 (5.9, 4.2), NF-κB (0.5, 4.0), E2F (5.7, 0.7), Myc (5.3, 2.0), ETS (inactive, 5.9), Notch (5.3, 4.6), Wnt (4.2, inactive) and Hedgehog (inactive, 5.0). Furthermore, a combination study between machaeriol C (**3**) and machaeridiol B (**6**) displayed additive effects for E2F, ETS, Wnt and Hedgehog pathways, where these compounds individually were either minimally active or inactive. None of the compounds inhibited luciferase expression driven by the minimal thymidine kinase promoter (pTK), indicating the lack of general cytotoxicity for luciferase enzyme inhibition at the 50 µM concentration in both of these cell lines. The significance of the inhibition of these signaling pathways via machaeridiol **5**–**7** and their cross-talk potential has been discussed.

## 1. Introduction

The *Machaerium* (Fabaceae) genus consists of approximately 130 species distributed in the tropical Americas [1]. It is a genus of shrubs or lianas and small- to medium-sized trees occurring throughout southern Mexico to Brazil and northern Argentina and Peru [2,3,4]. Specimens of this plant were collected from sandy soils in open forests in Maynas province, near Loreto, Peru, at an altitude of about 140–160 m. Several species of this genus are used in traditional medicines and are considered to have multiple medicinal properties, including antitussive, and the sap is used to cure aphthous ulcers of the mouth [4]. *M. floribundum* is used to treat diarrhea and menstrual cramps [4]. The presence of a wide array of secondary metabolites from *Machaerium*, including flavonoids, terpenoids and oxygenated phenolic compounds, together with their bioactivities, has recently been reviewed by Amen et al. [2]. Anticancer and antioxidative activities were reported from *M. cuspidatum* [5], and a cytotoxic cinnamylphenol was isolated from *M. aristulatum* [6].

We previously reported four unique (+)-*trans*-hexahydrodibenzopyran (HHDBP)- and three 5,6-*seco*-HHDBP-type phytocannabinoids, namely machaeriol A–D and machaeridiol A–C [7,8], respectively, as well as machaerifurogerol [9] from *Machaerium* Pers. (Rimachi, Y. 12161). We observed that machaeriols and hexahydrocannabinol share a structural similarity with the HHDBP nucleus, and machaeridiols and dihydrocannabidiol share a structural similarity with the 5,6-*seco*-HHDBP nucleus. These are a unique and rare aralkyl class of phytocannabinoids from a higher plant, analogous to cannabinoids from *Cannabis*. The absolute configuration of machaeriol (**1**–**4**) (Figure 1) was determined as 6a*S*,9*S*,10a*S* using [*α*]_D_, NOESY and CD spectra. Conspicuously, ∆^9^-tetrahydrocannabinol (THC; **8**), with its 10a*R* absolute configuration, displays a near mirror-image relationship with those of **1**–**4**. On the other hand, the configuration of machaeridiol (**5**–**7**) at 1*S*,3*S*,4*S* positions was found to be opposite to the configuration of cannabidiol (CBD; **9**) at 3*R*,4*R* positions. Compounds **1**–**7** were identified as HHDBP- and 5,6-*seco*-HHDBP-type “pharmacophores” for anti-infective natural products [7,8,9]. The machaeriols and machaeridiols have displayed potent antimicrobial activity, especially against methicillin-resistant *S. aureus* (MRSA), as well as inhibitory activities against malaria and leishmania [7,8,9]. Additionally, machaeriol A (**1**) and B (**2**) were evaluated against membranes purified from HEK-293 cells transfected with a recombinant plasmid encoding cannabinoid receptor 1 (CB1 receptor) using a radioligand binding assay. At concentrations of 1.0 and 0.1 μM, they displayed insignificant binding against CB1 receptor with affinity of between 7% and 10% for **1** and −3% and 3% for **2**, respectively [7].

The anticancer potential of the endocannabinoid system (ECS) via modulating pathways involved in cell growth, differentiation, migration and angiogenesis has been explored [10]. The role of the ECS in the pathogenesis of cancer has been well documented in the literature on key signaling pathways modulated by cannabinoids [10]. In 2017, Fonseca et al. reported that “both CB1 and CB2 are seven-transmembrane domain receptors coupled to G_i/o_ protein and their activation triggers several cancer pathways, including antiproliferative and pro-apoptotic effects attributed to activation of the alpha subunit of G_i/o_ that leads to inhibition of adenylate cyclase, which attenuate cyclic adenosine monophosphate (cAMP) synthesis and protein kinase A (PKA) activity, with the downregulation of gene transcription [10]. On the other hand, antiproliferative and pro-apoptotic effects of both CB1 and CB2 agonists have been attributed also to their ability to increase the synthesis of the pro-apoptotic sphingolipid ceramide” [11]. In addition, Thapa et al. (2011) reported that novel hexahydrocannabinol and machaeriol analogs showed potential anti-cancer activities involving the inhibition of cell proliferation and tumor angiogenesis [12].

The significance of the chemistry and bioactivity of these aralkyl phytocannabinoids **1**–**7**, isolated from the EtOH extracts of the stem bark, led to the examination of their binding affinities for CB1, CB2 and opioid (*κ*, *δ* and *µ*) receptors, using radioligand assays, together with their cytotoxic activities and modulation towards cancer-related signaling pathways. Cannabinoid and transcription factor assays have revealed that machaeridiol **5**–**7** show selective binding affinities for CB2 receptors, rather than CB1, and the inhibition of certain signaling pathways when compared to machaeriol (**1**–**4**) compounds. Their selectivity, specificity and cross-talk between cancer signaling was further evaluated via transcription factor multiplex assays.

## 2. Results and Discussion

### 2.1. Isolation of ***1***–***7***

Machaeriols (**1**–**4**) and machaeridiols (**5**–**7**) (Figure 1) were isolated from *n*-hexane and dichloromethane (DCM) partitions of ethanol (EtOH) extract of *Machaerium* Pers. Full details of their isolation and purification, using centrifugal preparative thin-layer chromatography (CPTLC) and high-pressure liquid chromatography (HPLC), have previously been reported [7,8,9]. In the current study, our focus was to evaluate the binding affinities of isolated compounds towards cannabinoid and opioid receptors, to evaluate their anticancer efficacy using cytotoxicity against solid tumor cells and to examine their selective modulation for cancer-related signaling pathways.

### 2.2. Cannabinoid and Opioid Receptor Affinity

Cannabinoid and opioid receptor affinities were determined via radioligand binding assays. The *n*-hexane and DCM partitions of ethanol extract showed an enhancement of affinities (>60%) for CB2 and *δ* and *μ* opioid receptors, while they were inactive against the CB1 receptor (Table 1). All isolated compounds were evaluated in receptor displacement assays, and none had any affinity for CB1. This lack of affinity to the CB1 receptor suggests that none of the compounds would be psychoactive (at least through the cannabinoid receptor), and psychoactivity for these compounds has not been reported in the literature. Machaeriol D (**4**), the hydroxylated analog of **2**, showed some affinity for CB2 with a *K*_i_ value of >1.3 * μM (Table 2). It showed little or no affinity for CB1 and opioid receptors, suggesting **4** has some selectivity for the CB2 receptor (Table 2). On the other hand, the 5,6-*seco*-HHDBPs, namely machaeridiol A (**5**), B (**6**) and C (**7**), showed affinities for the CB2 receptor (*K*_i_: >1.77 *; >2.18 *, >1.11 * μM), similar to the affinity observed for **4**. Due to the restricted solubility of **3** and **5**–**7** at concentrations at 33.3 μM and above in the aqueous buffer, the exact *K*_i_ values and functional assay for these compounds for CB2 could not be measured. Machaeriol C (**3**) and machaeridiol A–C (**5**–**7**) showed affinities for CB2 receptors, while **1** and **2** were found to be inactive. Machaeriol D (**4**) and machaeridiol (**5**–**7**) demonstrated weak CB2 affinities. In addition, all compounds were found to be weakly active against opioid (*δ* and *μ*) receptors. Earlier, machaeriol A (**1**) and B (**2**) were evaluated against the human recombinant cannabinoid receptor CB1 [7] and displayed insignificant affinity against CB1. Interestingly, the CB2 affinities of **4**–**7** were found to be similar to those reported for cannabidiol (**9**) [13].

### 2.3. Cytotoxicities against Solid Tumor Cells

The isolated compounds (**1**–**7**) were tested for in vitro cytotoxicity against five human solid tumor cell lines (SK-MEL, KB, BT-549, SK-OV-3 and HeLa) and two non-cancerous kidney cell lines (LLC-PK1 and Vero) (Table 3). Machaeridiol B (**6**) showed weak activities against HeLa, SK-MEL, KB and BT-549 cells with IC_50_ values between 32 and 38 µg/mL (Table 3), while compounds **2**, **3**, **5** and **7** were found to be inactive at the highest tested concentration of 50 µg/mL against most of the cells. A combination of machaeriol C and machaeridiol C, **3** + **7** (1:1), showed weak additive activity with IC_50_ values between 27 and 38 µg/mL against HeLa, KB, BT-549 and SK-OV-3 cells, compared to the absence of activity when each tested individually at 50 µg/mL. On the other hand, a combination of **3** + **6** (1:1) did not show any additive effect in this cytotoxicity assay.

### 2.4. Modulation of Cancer-Related Signaling Pathways

The *n*-hexane, DCM, ethanol extracts, isolated compounds **1**–**7** and CBD (**9**) were tested against a panel of transcription factors, including Stat3, Smad2/3, Ap-1, NF-κB, E2F, Myc, Ets, Notch, FoxO, Wnt, Hedgehog and pTK, which assess the activity of cancer-related signaling pathways. A battery of luciferase reporter gene vectors (i.e., luciferase expression was driven by the binding of transcription factors to multiple copies of synthetic enhancers within each vector) revealed that machaeridiol A–C (**5**–**7**) were more active compared to machaeriol A–D (**1**–**4**) (Table 4). Complete dose–response curves were determined for each compound, which revealed that machaeridiol **6** displayed activities (IC_50_ in µM in HeLa cells) toward Stat3 (4.7), Smad2/3 (1.2), AP-1 (5.9), NF-κB (0.5), E2F (5.7), Myc (5.3), and Notch (5.3) and Wnt (4.2), while its activity was generally comparable with CBD (**9**; IC_50_ 3.0–6.1 μM). Machaeridiol B (**6**) was found to be more potent compared to its two structural analogs **5** and **7** at inhibiting the activation of these signaling pathways. Interestingly, compound **6** was found to be inactive against ETS and Hedgehog in HeLa cells, compared to **5**, **7** and **9**, while **9** was inactive against AP-1 and E2F, compared to **6**. The features for the activation of machaeriol **1**–**4** throughout the panel of reporter genes appeared to be very similar, except **3**, which lacked activity against Stat3 and Smad2/3 pathways in HeLa cells. At the tested concentrations, none of the compounds inhibited luciferase expression driven by the minimal thymidine kinase promoter (pTK), indicating the lack of general cytotoxicity or non-specific luciferase enzyme inhibition (Table 4). Since compound **6** was active in HeLa cells, we analyzed its activity and determined IC_50_ values (µM) in T98G glioblastoma cells against these pathways, viz., Stat3 (1.4), Smad2/3 (3.0), AP-1 (4.2), NF-κB (4.0), E2F (0.7), Myc (2.0), and Notch (4.6) and Hedgehog (5.0). Although the inhibition of compound **6** across the panel in T98G cells was similar to HeLa cells, we found some interesting differences in activity. Compound **6** inhibited the Wnt pathway in HeLa cells (IC_50_ 4.2 µM), while it was inactive in T98G cells at the tested concentrations. In contrast, it was inactive against the ETS and Hedgehog pathways in HeLa cells, while it was active in T98G cells with IC_50_ values of 5.9 and 5.0 µM, respectively, for these pathways.

Amongst the tested compounds, machaeridiol B (**6**) was selected for further evaluation for a combination effect with machaeriol C (**3**) against these signaling pathways. Several different combinations of concentrations of compound **3** and **6** were tested to evaluate their additive effects (data not shown). At the concentration of 5 µM of compound **3**, the IC_50_ values of compound **6** were lowered for several pathways, notably AP-1, E2F, ETS, Notch, Wnt and Hedgehog. It is interesting to note that the IC_50_ values of compound **6** for the E2F and Wnt pathways were 5.7 and 4.2 µM, but they were more than halved to 2.5 and 1.1 µM for the combination of **3** + **6**, respectively. Likewise, **3** + **6** was active against the ETS and Hedgehog pathways (IC_50_ 3.0 and 4.0 µM, respectively), while **6** was inactive. These results clearly show the additive effect of **3** + **6** when tested in combination against these pathways. Moreover, the additive effect was visualized from the percentage (%) of inhibition against all signaling pathways (Table 5), where machaeriol C (**3**) exhibited a clear additive effect with **6** at a 10:2 (µM) ratio by inhibiting the Stat3, Smad2/3, AP-1, NF-κB, E2F, myc, Notch and Wnt pathways, compared to **3** and **6**, when tested individually at the same concentration. On the other hand, no inhibition was noted for pTK for **3**, **6** and **3** + **6**.

We previously reported the synthesis of a series of machaeridiol A and C analogs and their selective activity against a panel of cancer-related signaling transduction pathways using T98G *glioblastoma multiforme* cells [14]. Among these analogs, derivatives of pinosylvin **12** and **13** exhibited strong activity against many signaling pathways, with IC_50_ values between 1.86 and 4.45 μM against the Stat, Smad, Myc, AP-1, NF-kB, ETS and Notch pathways (Table 4). The activities of synthetic analogs **12** and **13** were in close agreement with the natural machaeridiols **5**–**7**, although they were tested in two different cells: T98G (glioblastoma) and HeLa (cervical) cells, respectively.

## 3. Experimental Section

### 3.1. Plant Material and Isolation of Compounds

The stem bark of *Machaerium* Pers. (Manuel Rimachi Y.—12161-MISS), previously identified as *M. multiflorum* Spruce. by late Professor Sydney T. McDaniel, was collected in 12 November 1997 from an open sandy forest near Loreto (Maynas), Peru. The voucher specimen (Manuel Rimachi Y. 12161) was deposited at Missouri Botanical Garden (https://tropicos.org/specimen/100326687; accessed 16 May 2023) [7,8,9]. The powdered stem bark of *Machaerium* Pers. was extracted with percolation via 95% EtOH, followed by the partitioning of concentrated EtOH extract with n-hexane and then with DCM. The bioassay-guided isolation was accomplished via the fractionation of n-hexane and DCM, and the active fractions were purified using CPTLC and HPLC, which yielded compounds **1**–**7**. Full details of the procedures were reported previously [7,8,9], and samples of these compounds are available in our lab.

### 3.2. Cannabinoid and Opioid Receptor Radioligand Binding Assay

#### 3.2.1. Reagents

CP55,940 was purchased from Tocris Bioscience (Minneapolis, MN, USA). BSA, Trizma^TM^ hydrochloride (Tris-HCl), penicillin, streptomycin and trypsin-EDTA were purchased from Sigma-Aldrich (St. Louis, MO, USA). Radioligands, GF/C, GF/B 96-well plates and MicroScint^TM^-20 were purchased from PerkinElmer (Waltham, MA, USA). Membrane preparation was carried out using a 50 mM Tris-HCl buffer with pH 7.4. 

#### 3.2.2. Cell Culture

Human embryonic kidney 293 (HEK293) cells (ATCC) were stably transfected with cannabinoid receptor subtypes 1 and 2 and maintained and harvested as described [15,16]. HEK293 cells stably transfected with *δ*, *κ* and *μ* opioid subtypes were a generous gift from Roth Laboratories (University of North Carolina at Chapel Hill, N.C., USA). CB and opioid cells were maintained as previously described [15,16].

#### 3.2.3. Membrane Preparation

Membranes were purified by first washing the cells with cold PBS and scraping the cells into cold 50 mM Tris-HCl, pH 7.4 buffer, which was centrifuged at 5200× *g* for 10 min at 4 °C; next, the supernatant was discarded, and the pellet was washed with more Tris-HCl buffer, homogenized via a Sonic Dismembrator (Fisher Scientific, Pittsburgh, PA, USA) and then centrifuged at 24,000× *g* for 40 min at 4 °C. Finally, the pellet was re-suspended in cold 50 mM Tris-HCl buffer, aliquoted into 2 mL vials and stored at −80 °C. The total membrane protein concentration was measured using a Pierce BCA Protein Assay Kit (Thermo Scientific, Rockford, IL, USA) as per the manufacturer’s protocol.

#### 3.2.4. Competitive Radioligand Binding Assays

Cannabinoid and opioid competitive radioligand binding assays were performed as previously described [15,16,17]. Saturation experiments were performed for all the receptors to determine the receptor concentration and radioligand dissociation constant (K_d_) for the membrane. Percent displacements were evaluated for all the cannabinoid and opioid subtypes with a triplicate of a fixed concentration (10 µg/mL for extracts and fractions and 10 μM for purified compounds). The samples competed with a tritium-labeled ligand with a known affinity of the receptor of interest {[^3^H]-CP55,940 for CB1R and CB2R, [^3^H]-U-69,593 for *κ*, [^3^H]-DAMGO for *μ*, or [^3^H]-enkephalin (DPDPE) for *δ*}, with the radioligand concentration equal to its K_d_. Control/test compounds were dissolved in DMSO at 10 μg/mL for extracts and fractions and 10 μM for purified compounds. Dilutions of the membrane, radioligand and control/test compounds were made in a Tris-EDTA buffer (50 mM Tris-HCl (pH 7.4), 20 mM EDTA, 154 mM NaCl and 0.2% fatty-acid free BSA), with pH = 7.4 for cannabinoids and 50 mM Tris-HCl (pH 7.4) for opioids. The percent displacement [18] was calculated to represent the ability of the samples to displace the radioligand binding for a given cannabinoid or opioid receptor subtype.

The percent displacement was calculated as follows:$$100-\frac{\mathrm{binding}\;\mathrm{of}\;\mathrm{compound}-\mathrm{nonspecific}\;\mathrm{binding}}{\mathrm{specific}\;\mathrm{binding}}\times 100$$

The competitive binding assays were performed using 12 serial dilutions of each compound ranging from 0.002 to 300 μM (control compounds were serially diluted from 10 μM to 0.06 nM). The cannabinoid assays were incubated for 90 min at 37 °C with gentle agitation. The opioid assays were incubated for 60 min at room temperature. Bound radioligands were collected on GF/C (cannabinoid) or GF/B plates (opioid) and washed 10 times with ice-cold 50 mM Tris-HCl (pH 7.4)/0.1% BSA (cannabinoid) or ice-cold 50 mM Tris-HCl (pH 7.4) (opioid). Radiodetection was measured with 50 µL (cannabinoid) or 25 µL (opioid) of MicroScint^TM^-20 on a TopCount NXT HTS Microplate Scintillation Counter (PerkinElmer, Waltham, MA, USA). The IC_50_ and *K*_i_ values were calculated via a non-linear curve fit model using GraphPad Prism 5.0 software (GraphPad Software, Inc., San Diego, CA, USA). Each compound was tested in triplicate unless stated otherwise.

### 3.3. Cytotoxicity Assays

In vitro cytotoxic activity was determined against five human cancer cell lines—malignant melanoma (SK-MEL), epidermal carcinoma oral (KB), ductal carcinoma breast (BT-549), ovary carcinoma (SKOV-3) and cervical (HeLa) cancer cells—and two non-cancerous kidney cell lines (LLC-PK_1_ and Vero). All cell lines were obtained from the American Type Culture Collection (ATCC, Rockville, MD, USA). Each assay was performed in 96-well tissue culture-treated microplates. The cells were seeded at a density of 25,000 cells/well and incubated for 24 h. Samples at different concentrations were added, and the cells were incubated again for 48 h. At the end of the incubation, the cell viability was determined using Neutral Red dye according to a modification of the procedure used by Borenfreund et al. [19]. IC_50_ values were determined from dose–response curves of the percent growth inhibition against test concentrations. Doxorubicin was used as a positive control, while DMSO was used as the negative (vehicle) control.

### 3.4. Transfection and Luciferase Assays

Hela cells (or T98G cells) from ATCC were plated in white opaque 384-well plates at a density of 4300 cells/well in 30 µL of growth medium (DMEM with 10% FBS and 1% Pen/Strep). The next day, the medium was aspirated and replaced with DMEM containing 1% FBS. The cells were transfected with respective plasmids using X-tremeGENE HP DNA transfection reagent (Roche). After 24 h of transfection, the test agents were added to the transfected cells, followed 30 min later by an inducing agent: interleukin 6 (IL-6) for Stat3, TGF-β for Smad, m-wnt3a for Wnt and PMA for AP-1, NF-κB, E2F, Myc, ETS, Notch and Hedgehog. No inducer was added for FoXO and pTK. After 4 h or 6 h of induction, the cells were lysed via the addition of the One-Glo luciferase assay system (Promega; Madison, WI). The light output was detected in a GloMax^®^-Multi+ Detection System with Instinct^®^ Software (Promega E 8051). This luciferase assay determines if the test agent is able to inhibit the activation of cancer-related signaling pathways. In the case of FoxO, the enhanced luciferase activity via the test agent was assessed [20].

## 4. Conclusions

The aralkyl phytocannabinoids machaeriol **1**–**4** and machaeridiol **5**–**7** are natural structural analogs of hexahydrocannabinol and dihydrocannabidiol, respectively. The only other bibenzyl analog of Δ^9^-THC, perrottetinene, was previously reported from the liverwort *Radula perrottetii* [21]. In the present report, a full set of compounds (**1**–**7**) was evaluated for CB1, CB2 and opioid receptor affinities and were found to be inactive against the CB1 receptor. Although their structures are analogous to psychoactive hexahydrocannabinol, machaeriols and machaeridiols showed no affinity for the CB1 receptor. Among the isolated compounds, machaeriol D (**4**) and machaeridiol A–C (**5**–**7**) showed affinities for the CB2 receptor (Table 2), similar to that reported for cannabidiol (**9**) [13]. The role of the ECS in the pathogenesis of cancer has been well documented [10]. Due to CBD’s and machaeridiols’ close structural similarities and their similar affinities for CB2, machaeridiol A–C (**5**–**7**) were examined against a panel of cancer-related signaling pathways (Table 3 and Table 4). CBD has recently received wide attention as a potential anticancer agent, in addition to its neuropharmacological effects. It suppresses the development of cancer in both in vitro (cancer cell culture) and in vivo (xenografts in immunodeficient mice) models [22]. Solinas et al. [23] reported that CBD down-regulated the ERK and AKT signaling pathways in glioblastoma cells (U87-MG and T98G) in a dose-dependent manner and also decreased the hypoxia-inducible factor (HIF-1a) expression in U87-MG cells. The effect of CBD on anti-inflammatory pathways, most significantly the Smad and NF-κB pathways, has been reported [24]. In addition, Freudisperger et al. [25] reported the functional cross-talk signaling between the TGF-β-dependent Smad and NF-κB pathways in tumor cells. Compounds **5**–**7**, particularly machaeridiol B (**6**), showed the specific inhibition of the Smad and NF-κB pathways, which further confirms the cross-talk signaling of these pathways, and they could serve as effective targets during drug development. In our previous report on machaeridiol analogs (14), both **12** and **13** exhibited the simultaneous modulation of multiple pathways, as observed by **5**–**7**, indicating compound **6** could be a potential lead molecule for targeted and combination therapies. The cross-talk between the other pathways in response to compound **6** reflects functional interactions and dependencies.

This appears to be the first comprehensive report on cannabinoid and opioid receptor affinities, cytotoxic activities and the modulation of cancer-related signaling pathways of natural compounds for machaeriol A–D (**1**–**4**) and machaeridiol A–C (**5**–**7**). Based on the foregoing discussion, further detailed evaluations of the aralkyl-phytocannabinoid-type machaeridiol A–C are required [8,9], and their previously reported synthetic analogs **12** and **13** [14] should undergo further detailed evaluation as potential anticancer leads, and effort should be focused on a comprehensive biological evaluation in cancer and normal cells and the establishment of their mechanism of action. The anticancer effects in animal models, combination studies with conventional anticancer drugs and pharmacokinetic–pharmacodynamic studies should be focused on for further development and lead optimization.

## Figures and Tables

**Figure 1 molecules-28-04162-f001:**
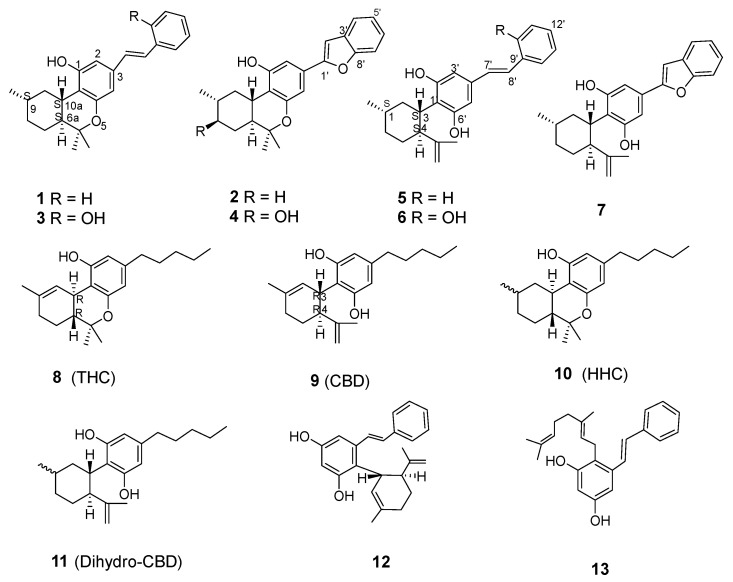
Chemical structures of compounds isolated from *Machaerium* Pers. (**1**–**7**), synthetic analogs (**12**, **13**) and related compounds (**8**–**11**).

**Table 1 molecules-28-04162-t001:** Percent displacement of *Machaerium* Pers. extract, fractions and compounds **1**–**7** from cannabinoid and opioid receptors.

Sample Name	Conc.	CB1	CB2	Δ	*ĸ*	*µ*
Mac-EtOH	10 μg/mL	16.7	4.1	45.7	3.4	-
Mac-Hex	10 μg/mL	58.5	63.7	52.2	26.2	64.5
Mac-DCM	10 μg/mL	-	-	68.2	20.3	65.5
Machaeriol A (**1**)	10 μM	18.4	24.4	1.8	-	52.9
Machaeriol B (**2**)	10 μM	-	13.7	61.7	-	37.6
Machaeriol C (**3**)	10 μM	43.6	55.5	4.6	6.4	-
Machaeriol D (**4**)	10 μM	4.5	81.5	28.2	-	16.8
Machaeridiol A (**5**)	10 μM	7.8	65.4	43.3	-	45.6
Machaeridiol B (**6**)	10 μM	45.2	56.8	50.2	-	27.6
Machaeridiol C (**7**)	10 μM	-	57.9	26.6	-	31.7
CP55,940	10 μM	87.8	101.8	-	-	-
Naloxone	10 μM	-	-	98.9	99.6	100.1

**Table 2 molecules-28-04162-t002:** *K*_i_ and p*K*_i_ values of compounds for CB2 and *δ* and *μ* opioid receptors.

Compound	CB2 *K*_i_ ± SEM (p*K*_i_), μM	Δ *K*_i_ ± SEM (p*K*_i_), μM	μ *K*_i_ ± SEM (p*K*_i_), μM
Machaeriol C (**3**)	>7.76 (<5.11) *	17.14 ± 2.61 (4.77)	NT
Machaeriol D (**4**)	>1.3 (<6.50) *	NA	NA
Machaeridiol A (**5**)	>1.77 (<5.75) *	9.41 ± 1.02 (5.03)	16.78 ± 3.31 (4.78)
Machaeridiol B (**6**)	>2.18 (<5.66) *	6.54 ± 0.77 (5.19)	4.73 ± 0.84 (5.33)
Machaeridiol C (**7**)	>1.11 (<5.95) *	12.19 ± 1.19 (4.91	4.01 ± 0.65 (5.40)
Naloxone (nM)	NT	25.93 ± 2.86 (7.59)	3.33 ± 0.38 8.48)
CP55,940 (nM)	0.924 ± 0.101 (9.03)	NT	NT

* Compounds were tested up to their solubility limit; conc. at 33.3 μM and above precipitated in aqueous assay buffer; NT = not tested; NA = not active.

**Table 3 molecules-28-04162-t003:** Cytotoxicity (IC_50_, µg/mL) * of compounds **2**–**7** and their various combinations.

Sample Code	SK-MEL	KB	BT-549	SK-OV-3	HeLa	LLC-PK1	Vero
**2**	>50	Na	Na	Na	Na	Na	Na
**3**	25	>50	>50	>50	Na	Na	Na
**5**	>50	>50	>50	Na	Na	Na	Na
**6**	32	38	36	>50	37	37	47
**7**	>50	>50	47	Na	Na	Na	Na
**3** + **5** (1:1)	>50	>50	37	>50	32	31	Na
**3** + **6** (1:1)	28	27	29	33	28	26	28
**3** + **7** (1:1)	27	34	32	38	32	31	Na
**5** + **7** (1:1)	50	30	35	Na	45	Na	Na
Doxorubicin	2.2	2.4	2.4	2	4.3	1.1	>5

* Test conc. 50, 16.7, 5.6 µg/mL; Na = not active up to 50 µg/mL; SK-MEL = human malignant melanoma, KB = Human Epidermal Carcinoma, Oral; BT-549 = ductal carcinoma, breast; SK-OV-3 = human ovary carcinoma; LLC-PK_1_ = pig kidney epithelial cells; VERO = monkey kidney fibroblasts. Compound **1** was inactive against all cell lines.

**Table 4 molecules-28-04162-t004:** Activity of compounds **1**–**7**, **9**, **12** and **13** and combination of **3** + **6** (IC_50_ in µM) ^a^ against cancer-related signaling pathways.

	Stat3(IL6)	Smad2/3(TGF-beta)	Ap-1(PMA)	NF-κB(PMA)	E2F(PMA)	Myc(PMA)	ETS(PMA)	Notch (PMA)	FoxO(None)	Wnt(m-wnt3a)	Hedgehog(PMA)
**1**	14.0	13.7	12.0	10.7	11.0	8.5	8.5	7.0	-	11.9	8.2
**2**	11.3	11.0	10.0	8.4	8.8	7.7	9.5	7.3	-	9.1	6.9
**3**	38.0	37.0	12.0	11.0	14.0	10.0	15.0	-	-	22.0	10.0
**4**	22.0	11.0	21.0	20.0	24.0	18.0	25.0	25.0	-	20.0	21.0
**5**	10.5	4.5	11.0	8.1	10.0	9.5	11.8	-	-	3.5	11.0
**6**	4.7	1.2	5.9	0.5	5.7	5.3	-	5.3	-	4.2	-
**7**	11.2	2.5	6.7	2.0	8.7	6.8	7.2	12	-	-	8.8
**9**	5.9	4.1	-	3.0	-	3.8	5.5	4.8	-	5.7	6.1
**6** + **3** *	3.8	1.2	3.5	0.9	2.5	4.3	3.0	3.8	-	1.1	4.0
**12** **	4.9	3.5	10.2	7.7	4.5	5.7	7.3	<5	-	12.1	7.0
**13** **	4.3	2.2	4.4	4.8	4.4	1.9	4.7	2.1	13.7	9.1	8.3
**6** 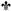	1.4	3.0	4.2	4.0	0.7	2.0	5.9	4.6	-	-	5.0

^a^ Values (from two independent experiments) are the IC_50_ or lowest concentration (both in µM) that maximally inhibited luciferase induction by 50–60%. Dash indicates that luciferase induction was not inhibited by more than 40% at 50 µmol/L. Compounds (at various concentrations) were added to cells 30 min before the addition of the indicated inducer in parenthesis and were harvested for luciferase assay four or six (Notch, FoxO, Wnt and Hedgehog) hours later. Compounds **1**–**7** and **9** were assayed in HeLa cells. No inducer was added to cells transfected with FoxO or pTK control vector. pTK was not inhibited (results not shown). * IC_50_ values of compound **6** when added in combination with compound **3** at the concentration of 5 µM. ** Data for synthetic analogs in T98G *glioblastoma multiforme* cells previously published [14] are included for comparison. 
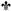
 Data of **6** in T98G cells. -: No activity detected.

**Table 5 molecules-28-04162-t005:** Additive effect (in % inhibition) * of the combination of compounds **6** and **3**.

Compound	Stat3 (IL-6)	Smad2/3 (TGF-β)	AP-1 (PMA)	NF-κB (PMA)	E2f (PMA)	Myc (PMA)	ETS (PMA)	Notch- (PMA)	FoxO (None)	Wnt (m-wnt3a)	Hedgehog (PMA)
**6** (2 µM)	82	109	70	76	80	111	107	103	105	113	92
**3** (10 µM)	92	93	61	70	50	52	66	115	120	81	66
**6** (2 µM) + **3** (10 µM)	42	30	36	33	42	53	41	71	152	46	53

* Values are percentages of luciferase induction at the given concentrations by the indicated inducers when compared to pTK as control. Test agents were added to the cells 30 min before the indicated inducer and were harvested for the luciferase assay four or six (Notch, FoxO, Wnt and Hedgehog) hours later. No inducer was added to the cells transfected with control vector (pTK) and FoxO.

## Data Availability

Data relevant to the paper will be available from the authors upon request.

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
