# Peer review of "Cannabinoid and Opioid Receptor Affinity and Modulation of Cancer-Related Signaling Pathways of Machaeriols and Machaeridiols from Machaerium Pers."

_molecules, 2023, doi:10.3390/molecules28104162_

Round 1

Reviewer 1 Report

The authors evaluated the effects of Machaerium Pers. compounds, namely machaeriol A-D (1-4) and machaeridiol A-C (5-7), against cannabinoid (CB1 and CB2) and opioid (, δ and μ) receptors. The study is well-designed and represented.

Author Response

Comment: The authors evaluated the effects of Machaerium Pers. compounds, namely machaeriol A-D (1-4) and machaeridiol A-C (5-7), against cannabinoid (CB1 and CB2) and opioid (k, δ and μ) receptors. The study is well-designed and represented.

Response: Many thanks for your generous comments.

Reviewer 2 Report

In this article entitled (Cannabinoid and Opioid Receptor Affinity and Modulation of Cancer-Related Signaling Pathways of Machaeriols and Machaeridiols from Machaerium Pers), the authors studied The significance of the chemistry and bioactivity of these aralkyl phytocannabinoids -7, isolated from the EtOH extracts of the stem bark, has led to the examination for their binding affinities for CB1, CB2 and opioid (k, δ and µ) receptors, using radioligand assays, together with their cytotoxic activities and modulation towards cancer-related signaling pathways.

The manuscript is interesting however, some aspects should be better explored and explained before acceptance for publication.

Comments

1-    The manuscript needs revision regarding typos errors and grammar mistakes.  

2-    Authors should revise the author's guidelines regarding the format of tables and their citations in the text.

3-    The conclusion is too long

4-    The results need interpretation with previous findings

Minor editing of English language is required 

Author Response

Comment: In this article entitled (Cannabinoid and Opioid Receptor Affinity and Modulation of Cancer-Related Signaling Pathways of Machaeriols and Machaeridiols from Machaerium Pers), the authors studied The significance of the chemistry and bioactivity of these aralkyl phytocannabinoids -7, isolated from the EtOH extracts of the stem bark, has led to the examination for their binding affinities for CB1, CB2 and opioid (k, δ and µ) receptors, using radioligand assays, together with their cytotoxic activities and modulation towards cancer-related signaling pathways. 

The manuscript is interesting however, some aspects should be better explored and explained before acceptance for publication.

  1. The manuscript needs revision regarding typos errors and grammar mistakes.

Response: Thank you for the suggestion. The manuscript is thoroughly revised now. All edits are done using track-change mode.

  1. Authors should revise the author's guidelines regarding the format of tables and their citations in the text.

Response: We have formatted the manuscript where necessary, and tables are formatted correctly according to the guidelines. We have rechecked it and all looks good to us. Table(s) citations are included appropriately in the text.

  1. The conclusion is too long.

Response: The Conclusion Section is now reduced carefully by removing texts relevant to interpretation of results to Results and Discussion Section (see below, item 4) and deleting repetitions from Conclusion Section, so that the significance and purpose of our work can be maintained.

  1. The results need interpretation with previous findings.

We have now moved texts relevant to interpretation of results with previous findings from conclusion section to Results and Discussion Section (see highlighted text). The closest previous findings are the work on CBD and machaeridiol synthetic analogs, which are already included/ cited (refs 13 1nd 14). Due to the movement of texts, references are now adjusted in the text and Reference Section.

  1. Minor editing of English language is required.

Response: The manuscript is edited by one of our co-authors, Ms. Janet Lambert, who is a native English speaker. 

Reviewer 3 Report

The author said that Compounds 1-7 were also evaluated for cytotoxicity against human cancer cells SK-MEL, KB, BT-549, SKOV-3, and HeLa and two non-cancerous kidney cell lines (LLC-PK1 and Vero) And This appears to be the first comprehensive report on cannabinoid and opioid receptor affinities, cytotoxic activities, and modulation of cancer related signaling pathways of natural compounds, machaeriol A-D (1-4) and machaeridiol A-C (5-7) .This paper covered many experiments and reaction of signalling and explain the mechanism.  This paper will be useful of research in anticancer drugs.

Author Response

Comment: The author said that Compounds 1-7 were also evaluated for cytotoxicity against human cancer cells SK-MEL, KB, BT-549, SKOV-3, and HeLa and two non-cancerous kidney cell lines (LLC-PK1 and Vero) And This appears to be the first comprehensive report on cannabinoid and opioid receptor affinities, cytotoxic activities, and modulation of cancer related signaling pathways of natural compounds, machaeriol A-D (1-4) and machaeridiol A-C (5-7) .This paper covered many experiments and reaction of signalling and explain the mechanism.  This paper will be useful of research in anticancer drugs.

Response: Many thanks for your generous comments.

Round 2

Reviewer 2 Report

Authors did all required revisions. The manuscript  is suitable now for publication in Molecules.